# Ergothioneine Attenuates Oxaliplatin-Induced Peripheral Neuropathy Without Compromising Antitumor Efficacy

**DOI:** 10.3390/ijms262110263

**Published:** 2025-10-22

**Authors:** Takumu Yamada, Takumi Iwasawa, Ryohma Tsuchiya, Tomoaki Ito, Kazunori Kato

**Affiliations:** 1Department of Biomedical Engineering, Graduate School of Science and Engineering, Toyo University, Saitama 350-8585, Japan; s46b02200046@toyo.jp (T.Y.);; 2Institution of Life Innovation Studies, Toyo University, Tokyo 115-8650, Japan; 3Shizuoka Medical Research Center for Disaster, Juntendo University Shizuoka Hospital, Shizuoka 410-2295, Japantomo-ito@juntendo.ac.jp (T.I.); 4Department of Surgery, Juntendo University Shizuoka Hospital, Juntendo University School of Medicine, Shizuoka 410-2295, Japan; 5Department of Nutrition Sciences, Graduate School of Health and Sports Sciences, Toyo University, Tokyo 115-8650, Japan

**Keywords:** oxaliplatin, ergothioneine, oxaliplatin-induced peripheral neuropathy (OIPN), colorectal cancer

## Abstract

Colorectal cancer remains one of the leading causes of cancer-related mortality both in Japan and worldwide. Oxaliplatin (L-OHP) is a key chemotherapeutic agent used in the treatment of colorectal and other malignancies; however, its clinical use is often limited by the development of oxaliplatin-induced peripheral neuropathy (OIPN). In this study, we investigated ergothioneine (EGT), a natural antioxidant abundant in mushrooms, for its potential to mitigate OIPN without compromising the antitumor efficacy of L-OHP. Using the SH-SY5Y neuroblastoma cell line and differentiated neurons, we assessed the effects of EGT on L-OHP-induced apoptosis, oxidative stress, and axonal degeneration. We further evaluated whether EGT interferes with the anticancer activity of L-OHP using cultured cancer cell lines and a tumor-bearing mouse model. EGT suppressed L-OHP-induced apoptosis in neuronal cells and preserved axonal structures in differentiated neurons. Importantly, EGT had no adverse effect on the antitumor efficacy of L-OHP, as evidenced by unchanged cancer cell proliferation, tumor volume, and body weight in treated mice. These findings suggest that EGT may be a promising adjuvant for preventing OIPN while maintaining the therapeutic benefits of L-OHP.

## 1. Introduction

Colorectal cancer, encompassing both colon and rectal malignancies, remains a major global health concern. In Japan, it ranks first in incidence and second in cancer-related mortality. Although mortality rates have declined since the late 1990s—largely due to improved screening programs and advancements in therapy—colorectal cancer remains the third most commonly diagnosed cancer in both the United States and worldwide [1,2]. Despite significant progress, the persistently high mortality associated with colorectal cancer highlights the continued need for more effective and safer therapeutic strategies.

Oxaliplatin (L-OHP), a platinum-based chemotherapeutic agent, plays a central role in colorectal cancer treatment. Platinum compounds are widely employed against various malignancies, including ovarian and breast cancers. However, their clinical utility is often restricted by severe adverse effects such as nephrotoxicity, myelosuppression, and, most notably, chemotherapy-induced peripheral neuropathy (CIPN) [3,4]. CIPN is a dose-limiting toxicity believed to result from the accumulation of platinum compounds in dorsal root ganglion (DRG) neurons via the organic cation/carnitine transporter 1 (OCTN1), leading to axonal damage and diverse sensory abnormalities [5,6]. Its incidence can reach up to 68% within the first month of treatment, and persistent symptoms can significantly impair long-term quality of life (QOL) [7,8]. CIPN is categorized into acute and chronic forms, each involving distinct mechanisms [8]. Acute CIPN is associated with dysregulation of ion channels such as Na^+^ and Ca^2+^ [9,10,11], as well as OCTN1-mediated transport [12], resulting in immediate pain and sensory disturbances. In contrast, chronic CIPN is linked to nuclear DNA damage [13,14] and mitochondrial dysfunction caused by oxidative stress [15,16]. Additionally, oxalate, a metabolic byproduct of L-OHP, may interfere with voltage-gated sodium channels, disrupt sodium currents, and exacerbate neurotoxicity [17]. To mitigate L-OHP-associated toxicities, combination regimens such as FOLFOX—which includes L-OHP, 5-fluorouracil (5-FU), and Leucovorin (I-LV)—have become the standard of care. Nonetheless, more than 80% of patients receiving FOLFOX still develop CIPN, underscoring the urgent need for novel adjunctive therapies [18].

Ergothioneine (EGT) is a sulfur-containing, histidine-derived natural product with potent antioxidant activity. Humans and other mammals cannot synthesize EGT; instead, it is obtained from the diet, particularly from edible mushrooms. EGT is widely distributed across human tissues, functioning as a potent antioxidant that safeguards cells from oxidative stress. Beyond this antioxidative role, accumulating evidence indicates that EGT exerts broad cytoprotective effects in diverse cell types, with its neuroprotective activity representing a notable aspect of this property. Such properties suggest potential clinical relevance, particularly in conditions where oxidative stress and neuronal damage are major contributors, including chemotherapy-induced toxicities and cognitive impairment.

Preclinical studies have shown that EGT can alleviate L-OHP-induced CIPN in animal models, likely through its antioxidant properties [19]. In addition to its neuroprotective potential, EGT exhibits selective cytotoxicity against colorectal cancer cell lines via reactive oxygen species (ROS)-mediated mechanisms [20]. Interestingly, the cell cycle arrest induced by EGT varies by cell type—causing S-phase arrest in HT-29 cells and G2/M-phase arrest in SW620 and LoVo cells—and appears to induce both necrosis and apoptosis. Notably, EGT is taken up into cells via OCTN1, the same transporter involved in L-OHP uptake [21], and does not exhibit cytotoxicity toward non-cancerous cell lines even at concentrations as high as 2 mM, suggesting a favorable therapeutic index [20].

Although clinical studies have not demonstrated a statistically significant association between circulating EGT levels and the severity of CIPN [22], preclinical evidence supports its potential role in mitigating neurotoxicity. However, the direct effects of EGT on human neuronal cells remain poorly understood. Moreover, while EGT’s cytotoxic effects on tumor cells have been established, its influence on the anticancer efficacy of L-OHP has not yet been systematically evaluated [23]. Therefore, this study aimed to investigate the dual role of EGT: first, to assess its neuroprotective effects against L-OHP-induced neurotoxicity in human neuronal cells; and second, to evaluate its impact on the therapeutic efficacy of L-OHP in colorectal cancer, both in vitro and in vivo. The findings of this study are expected to provide mechanistic insights into the potential application of EGT as an adjuvant agent to enhance both the safety and efficacy of platinum-based chemotherapy. metastatic capacity in a murine model.

## 2. Results and Discussion

### 2.1. EGT Protects SH-SY5Y Cells from Oxaliplatin-Induced Cytotoxicity via Anti-Apoptotic and Antioxidant Mechanism

Platinum-based chemotherapeutic agents are widely used across a broad range of malignancies. However, their most debilitating adverse effect, peripheral neuropathy, constitutes a dose-limiting toxicity because treatment often must be discontinued once it develops. In this study, we investigated the neuroprotective effect of EGT, an antioxidant transported via pathways shared with L-OHP, against L-OHP–induced neuronal injury. As a result, 48 h pretreatment of SH-SY5Y neuroblastoma cells with EGT followed by L-OHP treatment (final concentration, 10 μM) markedly increased cell viability to approximately 85%, compared with approximately 25% in cells not pretreated with EGT (Figure 1A). Microscopic analysis and flow cytometry analysis by annexin V/7-AAD staining further revealed that the increase in apoptotic cell proportion induced by L-OHP was significantly reduced by EGT co-treatment (Figure 1B,C), suggesting that EGT may protect against L-OHP-induced neurotoxicity by inhibiting apoptotic pathways.

Neurons are particularly susceptible to oxidative stress due to their high oxygen consumption and lipid-rich composition, which render them prone to lipid peroxidation [24]. Malondialdehyde (MDA), a byproduct of polyunsaturated fatty acid peroxidation, serves as a marker of oxidative stress-induced lipid peroxidation [25]. Therefore, we measured intracellular reactive oxygen species (ROS) and MDA levels as indicators of oxidative stress. As shown in Figure 2A,B, SH-SY5Y cells stained with DCFH-DA and MDA reagent exhibited increased total ROS accumulation and lipid peroxidation following treatment with L-OHP. EGT markedly reduced L-OHP-induced ROS accumulation and concomitantly decreased MDA levels, supporting its role in mitigating oxidative damage. We further examined the expression cleaved caspase-3 and cleaved poly ADP-ribose polymerase (PARP), both hallmark markers of apoptosis induced by L-OHP, by Western blotting. The increased levels of cleaved caspase-3 and PARP were notably suppressed by EGT co-treatment (Figure 3A,B).

Several studies have shown that EGT is a potent scavenger of ROS and protects normal cells from oxidative-stress–induced injury. Gokce et al. reported that EGT prevented HgCl2-induced endothelial dysfunction—a robust ROS-generating model—accompanied by significant reductions in intracellular ROS and MDA when cells exposed to HgCl2 were subsequently treated with EGT [26]. Consistent with these findings, we observed that EGT reduced L-OHP–induced increases in ROS and MDA. Ultraviolet (UV) radiation is another common source of oxidative stress in daily life; UV irradiation triggers ROS production and activation of cell-death signaling, leading to apoptosis of keratinocytes and fibroblasts. Ko et al. further showed that EGT protects keratinocytes by suppressing ROS production and decreasing levels of cleaved pro-apoptotic proteins [27]. Taken together, prior reports are consistent with our finding that EGT mitigates L-OHP–induced neuronal toxicity via dual antioxidant and anti-apoptotic mechanisms.

### 2.2. EGT Preserves Neuronal Function in Differentiated SH-SY5Y Cells Under Oxaliplatin-Induced Neurotoxicity

SH-SY5Y cells have a stable 47-chromosome karyotype and can be differentiated from a neuroblast-like state into mature human neurons using retinoic acid (RA), phorbol esters, or neurotrophins. Different protocols enrich distinct neuronal subtypes (adrenergic, cholinergic, dopaminergic), making SH-SY5Y a versatile neurobiology model. We then differentiated SH-SY5Y cells to evaluate whether EGT mitigates L-OHP–induced neurotoxicity in mature neurons. Using a standard retinoic acid protocol [23], we converted SH-SY5Y neuroblastoma cells from an epithelial-like morphology to a neurite-bearing neuronal phenotype with extensive neurite outgrowth (Figure 4A). Neuronal maturation was confirmed by immunofluorescence for NF-H on day 19 (Figure 4B). Using these differentiated neurons, we next tested the cytoprotective effect of EGT against L-OHP–induced injury. NF-H immunofluorescence showed that L-OHP reduced axon counts relative to control, whereas EGT co-treatment largely restored axon counts (Figure 5A). Quantitatively, L-OHP (2 μM) significantly decreased axon counts compared with control (mean ± SEM, 1.16 ± 0.22 vs. 2.71 ± 0.53 axons per cell; *n* = 9), while co-treatment with EGT (20 μg/mL) restored axon numbers toward control levels (2.06 ± 0.42; one-way ANOVA with Tukey’s post hoc test, *p* = 0.011 vs. L-OHP; Figure 5B). Similar to the prevention properties of EGT against L-OHP-induced neurotoxicity in undifferentiated neurons, EGT treatment increased neurite numbers per cell, suggesting promotion of axonal integrity under neurotoxic conditions in differentiated neurons.

In the present study, we did not directly investigate the mechanism by which EGT mitigates L-OHP-induced neurotoxicity. However, previous work by Nishida et al. demonstrated that OCTN1 and OCTN2 are expressed in DRG neurons and that substrates or inhibitors of these transporters can modulate L-OHP uptake within the neurons without altering plasma concentrations. These findings suggest a potential pathway through which EGT may reduce platinum accumulation in DRG neurons and alleviate neurotoxicity.

In addition to cardiovascular and skin damages protected by EGT, several reports have demonstrated that EGT exerts neuroprotective effects in various neurological diseases, including Alzheimer’s disease (AD) and Parkinson’s disease (PD). It has been known that amyloid-beta (Aβ) oligomers formed by Ab aggregation induce the hyperphosphorylation of tau protein and neurotoxicity. Shibagaki et al. reported that EGT suppress the increased expression of phosph-tau proteins involved in Aβ production, thereby mitigating neurotoxicity [28]. Furthermore, the accumulation of α-synuclein (α-syn) plays a key role in the pathogenesis of PD. Several reports demonstrated that EGT effectively inhibited α-syn aggregation, oxidative stress and associated cytotoxicity in vitro and in vivo [29]. Our results also show that EGT mitigates anticancer drug-induced neurotoxicity in both undifferentiated and differentiated neuronal cells in vitro and peripheral neuropathy in vivo rat model [19]. Notably, EGT is non-toxic to normal cells even at high concentrations [20], suggesting that higher doses than those used in this study may further enhance neuroprotection.

### 2.3. EGT Does Not Adversely Affect the Antitumor Effect of L-OHP

Previous reports and our own experiments demonstrate that EGT exerts potent cytoprotective effects, particularly protecting normal cells—including neurons—against oxidative stress. A remaining concern, however, is whether EGT might confer similar protection to cancer cells. If EGT were used to mitigate chemotherapy-induced adverse effects, any attenuation of antitumor efficacy would undermine its clinical utility. To address this, we evaluated whether EGT affects the antitumor activity of L-OHP using the colon cancer cell lines HCT116 and DLD-1 in vitro and in vivo. First, cells were treated with L-OHP with or without EGT, and cell proliferation was assessed by the alamarBlue assay. L-OHP effectively induced cell death in both cell lines, and EGT co-treatment had no significant effect on cell viability (Figure 6A,B). Consistent with L-OHP–mediated growth inhibition, the increases in cleaved caspase-3 and cleaved PARP were unaffected by EGT (Figure 6C–E). These results indicate that EGT does not compromise the antitumor efficacy of L-OHP and may selectively mitigate its neurotoxic adverse effects. Although EGT’s antioxidant activity raised concerns that it might attenuate ROS-mediated tumor cell killing, our findings did not support this. It is possible that low concentrations of EGT protect neurons without interfering with L-OHP’s cytotoxic effects, whereas higher concentrations could potentially diminish ROS-dependent antitumor activity. The precise molecular mechanisms underlying EGT’s neuroprotective effects, as well as possible pharmacokinetic or pharmacodynamic interactions with L-OHP, warrant further investigation.

Furthermore, to investigate whether similar effects occur in vivo, we evaluated the in vivo effects of EGT on platinum-based chemotherapy. Nishida et al., rat study, reported that EGT can attenuate OIPN. However, this study did not address whether EGT influences the antitumor efficacy of L-OHP. In our study, we therefore adopted similar doses in the present experiments [19]. Nude mice (*n* = 6 per group) were subcutaneously inoculated with HCT116 colon cancer cells and received EGT on days 7, 8, 11, 12, 18, 19, 22, and 23 and FOLFOX (5-fluorouracil, leucovorin, and oxaliplatin) on days 8, 12, 19, and 23 after tumor inoculation (Figure 7A). Tumor volumes were significantly reduced in both the FOLFOX and FOLFOX plus EGT groups compared with the control and EGT-alone groups. As expected, there was no significant difference in tumor growth between the FOLFOX and FOLFOX plus EGT groups, indicating that EGT did not compromise the antitumor efficacy of FOLFOX (Figure 7B). Although body weight loss was observed in FOLFOX-treated mice, none reached the humane endpoint. Notably, FOLFOX plus EGT resulted in a slower rate of body weight loss than FOLFOX alone (Figure 7C). These findings suggest that EGT may mitigate chemotherapy-induced weight loss—a clinically relevant adverse effect of platinum-based regimens—while preserving the therapeutic benefit of oxaliplatin.

In our experiments, we have not examined the effects on neuroprotection in mice. This is partly because mouse DRGs are much smaller than rat DRGs, making historical evaluation technically challenging, whereas rat DRGs are relatively large and easier to analyze. Nevertheless, in a previous study, Nishida et al. examined the combined effects of EGT and L-OHP in the DRG of rats and demonstrated that EGT attenuated L-OHP-induced neuropathy [19]. In the present study, although female mice were used instead of rats, the dosing regimen was based on their report. We therefore consider it plausible that the protective effect of EGT against L-OHP-induced neuropathy observed in rats may also underlie the effects observed in our mouse model. Moreover, our study did not perform a behavioral test for neurotoxicity, such as the allodynia test. Nishida et al. reported that in rats, L-OHP administration increased the frequency of withdrawal responses to filament stimulation, which was significantly reduced when combined with EGT. Although we did not conduct the allodynia test in our mouse experiments, we followed the same dosing regimen and administration schedule as in the Nishida et al. study. Based on these conditions, it is therefore plausible that EGT may similarly mitigate L-OHP-induced mechanical hypersensitivity in mice.

Finally, to investigate whether transporter expression could influence the cellular response to EGT and L-OHP, we analyzed the expression levels of related transporters by qPCR. The uptake of EGT and L-OHP has previously been reported to involve *OCTN1* and *OCTN2* [30,31], while *ATP7a* and *ATP7b* are known as copper efflux transporters that also mediate the efflux of platinum compounds such as cisplatin and oxaliplatin [32]. Our qPCR analysis revealed that the expression levels of *ATP7a* and *ATP7b* were higher in SH-SY5Y cells compared with colorectal cancer cell lines (DLD-1 and HCT-116) (Figure 8A,B). *ATP7a* is broadly expressed in tissues beyond the intestinal epithelium, kidney, and liver [32,33]. Previous studies have reported that *ATP7a* mediates the efflux of platinum compounds, thereby reducing intracellular platinum accumulation [34]. At the same time, *ATP7a* expression has been observed in several malignant tumors and is associated with poor tumor response in patients [35]. Taken together, these findings suggest that *ATP7a* and *ATP7b* may contribute to the development of OIPN but could also reduce the antitumor efficacy of oxaliplatin, indicating that they may not represent ideal therapeutic targets. Nevertheless, in tumors lacking *ATP7a* expression, the combination of L-OHP and EGT may provide neuronal protection without impairing the antitumor effect of L-OHP.

In addition, cation transport mechanisms may play a crucial role. EGT is primarily trans-ported by *OCTN1* (*SLC22A4*) [36]. Indeed, studies using *OCTN1* knockout mice have shown that hepatic uptake of EGT is approximately 70-fold lower than in wild-type mice, while intestinal uptake is reduced about 14-fold, strongly indicating that *OCTN1* is a key mediator of EGT transport, including into neurons [37]. In contrast, oxaliplatin (L-OHP) uptake has been reported to involve several transporters, notably *OCT1, OCT2, OCTN1, and OCTN2* [30,31]. Moreover, Kevin et al. reported that L-OHP-induced neuropathy is associated with *OCT2* activity, and that pharmacological inhibition of *OCT2* can mitigate this adverse effect [38]. These findings raise the possibility that EGT may also interact with *OCT2*, thereby influencing drug distribution. However, our qPCR results showed that *OCTN1* was upregulated in SH-SY5Y cells, which are neurons, more than in DLD-1 and HCT-116 cells, which are colon cancer cells, as in previous studies, while *OCTN2* was downregulated in SH-SY5Y cells more than in DLD-1 cells (Figure 8C,D). Thus, it is clear that EGT acts on *OCTN1*, but its involvement with *OCTN2* remains to be examined. Nonetheless, differences in transporter expression between neuronal and tumor cells could consequently alter intracellular drug accumulation and ultimately impact both neurotoxicity and antitumor efficacy.

Previous studies have demonstrated that EGT administration in mice for approximately 40 days suppressed amyloid-β accumulation in the hippocampus and mitigated oxidative damage in brain tissue, thereby preventing amyloid- β–induced impairments in memory and learning [39]. Similarly, in models of metabolic dysfunction–associated liver disease, EGT reduced body weight, adiposity, and circulating lipid levels while enhancing autophagy, which was accompanied by significant reductions in oxidative stress, inflammation, and apoptosis [40]. Although typical human exposure is estimated at 5–10 mg/day, long-term administration at 1600 mg/kg body weight/day for 90 days in mice caused no adverse effects, supporting its safety as a dietary supplement [41]. Furthermore, synthetic L-EGT was approved as a novel food ingredient in 2016, and subsequent safety evaluations in 2017 confirmed that intake up to 800 mg/kg body weight/day is considered safe even for infants, pregnant women, and lactating women [42].

### 2.4. Limitations

This study has several limitations. First, we did not conduct histological evaluations of peripheral nerves in vivo, which would have been important to confirm the neuroprotective effects of EGT against OIPN. Second, our in vivo experiments did not include behavioral testing, such as the allodynia test. Although previous studies in rats have shown that EGT ameliorates L-OHP-induced hypersensitivity, whether a similar effect occurs in mice remains to be confirmed. Third, sex differences and DRGs were not addressed in our in vivo experiments. Pharmacokinetics and neurotoxicity of oxaliplatin can differ between male and female mice, which may influence both the extent of peripheral neuropathy and the protective effects of EGT. Future studies should include both sexes to determine whether the observed effects are consistent across male and female animals. Fourth, we did not assess mitochondrial function, oxidative stress responses, or inflammatory signaling pathways that may underlie the protective effects of EGT. OIPN has been reported to be closely associated with mitochondrial dysfunction and increased oxidative stress [19,43,44]. Previous studies have shown that antioxidant compounds can alleviate oxaliplatin-induced ROS accumulation and mitochondrial damage, thereby reducing OIPN in rats [19,45]. Therefore, given its strong antioxidant properties, EGT may protect neurons and mitigate OIPN by reducing mitochondrial dysfunction and ROS levels; however, this hypothesis remains to be verified experimentally.

### 2.5. Future Directions

To advance this preliminary work, we will: (1) define transporter-mediated uptake of EGT and oxaliplatin (OCTN1/OCTN2/OCT1/OCT2) using genetic perturbation and uptake assays; (2) Immunohistochemical analysis of OCTN1- or OCTN2-knockout mice to determine whether the neuroprotective effect of EGT is abolished.; (3) map mitochondrial and oxidative-stress pathways (Δψm, OCR/ECAR, Nrf2/HO-1, 4-HNE, 8-oxo-dG) and inflammatory signaling; (4) parse acute versus chronic CIPN by ion-channel measurements (NaV/CaV) and DNA-damage markers (γH2AX); (5) conduct in vivo, pair behavioral tests (von Frey, cold plate), nerve-conduction studies, and DRG/sciatic histology with dose/schedule optimization of EGT (pre- vs. co-administration); (6) ensure oncologic safety by expanding tumor models, including immunocompetent settings; and (7) evaluate sex differences. We are also designing an early-phase clinical study of EGT combined with oxaliplatin-based chemotherapy with predefined OIPN and pharmacokinetic endpoints. Taken together, these findings highlight EGT’s favorable safety profile and demonstrated efficacy in preclinical models, reinforcing the significance of our results and suggesting that they may serve as a foundation for future translational studies and early clinical applications in humans.

## 3. Materials and Methods

### 3.1. Cell Culture and Neuronal Differentiation

Immortalized SH-SY5Y neuroblastoma cells (purchased from KAC Co., Ltd., Kyoto, Japan) were used for differentiation into mature neurons. Cells were cultured in DMEM (Sigma-Aldrich, St. Louis, MO, USA) supplemented with 10% fetal bovine serum (FBS; Biowest, Riverside, MO, USA), 1% penicillin–streptomycin (Life Technologies Corporation, New York, NY, USA), and 1% GlutaMAX (Life Technologies Corporation) at 37 °C in 5% CO_2_. Neuronal differentiation was induced following a standard retinoic acid–based protocol as described by Shipley et al. [23] with minor modifications.

HCT-116 and DLD-1 Human colorectal cancer cell lines, HCT116 (obtained from RIKEN Cell Bank, Ibaraki, Japan) and DLD-1 (obtained from JCRB Cell Bank, Osaka, Japan), were cultured in RPMI-1640 (Sigma-Aldrich) supplemented with 10% FBS, 1% GlutaMAX, 1% penicillin–streptomycin, and sodium pyruvate [final concentration per manufacturer’s recommendation] at 37 °C in 5% CO_2_. All procedures were performed under aseptic conditions in a Class II biosafety cabinet.

### 3.2. Sulforhodamine B (SRB) Cell Survival Assay

Cell viability was assessed using the sulforhodamine B (SRB) assay, as previously described [46]. Cells were treated with compounds for the indicated durations. Cells were then fixed by adding 50 μL/well of 50% (*w*/*v*) trichloroacetic acid (TCA) and incubated at 4 °C for 1 h. The supernatant was discarded, and wells were washed three times with Milli-Q water. After air-drying briefly, 50 μL/well of 0.4% (*w*/*v*) sulforhodamine B (SRB) solution was added and incubated for 10 min at room temperature (RT). Unbound dye was removed by washing three times with 1% (*v*/*v*) acetic acid (FUJIFILM Wako Pure Chemical Corporation, Tokyo, Japan), and plates were air-dried overnight. Bound dye was solubilized with 150 μL/well of 10 mM Tris base (FUJIFILM Wako) on a microplate shaker (Atto Corporation, Tokyo, Japan) for ~10 min. Absorbance was measured at 525 nm using a SpectraMax M5 microplate reader (Molecular Devices, San Jose, CA, USA).

### 3.3. Annexin V–FITC/7-AAD Apoptosis Assay

Cells were seeded at 10,000 cells/well into 6-well plates (Corning Incorporated, Corning, NY, USA) and cultured under the same conditions as above. Cells were harvested and resuspended in binding buffer, and staining was performed using the ApoScreen^®^ Annexin V Apoptosis Kit-FITC (SouthernBiotech, Birmingham, AL, USA) together with 7-aminoactinomycin D (7-AAD; Sigma-Aldrich), according to the manufacturers’ instructions. For permeabilization where indicated, 0.1% Triton X-100 in PBS (FUJIFILM Wako) was used. Samples were analyzed on a BD FACSymphony A1 flow cytometer (Becton, Dickinson and Company, Franklin Lakes, NJ, USA).

### 3.4. Malondialdehyde (MDA) Assay

Lipid peroxidation was quantified using a malondialdehyde (MDA) assay kit (DOJINDO LABORATORIES, Kumamoto, Japan) according to the manufacturer’s protocol. Absorbance was read on a SpectraMax M5 microplate reader (Molecular Devices).

### 3.5. Reactive Oxygen Species (ROS) Assay

Intracellular ROS levels were measured using the Total ROS Detection Kit (DOJINDO LABORATORIES) per the manufacturer’s instructions. Fluorescence was acquired on a BD FACSymphony A1 flow cytometer (Becton Dickinson, Franklin Lakes, NJ, USA) and analyzed with FlowJo software (version 10.10.0, Becton Dickinson).

### 3.6. Western Blotting

Cells were seeded into 100 mm dishes at 5 × 10^5^ cells per dish and allowed to adhere overnight. Ergothioneine (EGT) was added to a final concentration of 15 μg/mL for 48 h, followed by oxaliplatin (L-OHP) at 2 μM for an additional 24 h. After treatment, cells were collected and lysed in RIPA buffer (Table 1) to extract total protein. Equal amounts of protein were separated by SDS-PAGE and transferred to polyvinylidene difluoride (PVDF) membranes. Membranes were blocked in 5% non-fat dry milk in PBS-T (PBS with 0.05% Tween-20), incubated with primary antibodies, and then with horseradish peroxidase (HRP)–conjugated secondary antibodies.

Primary antibodies: rabbit anti-cleaved caspase-3 (Cell Signaling Technology, Danvers, MA, USA), rabbit anti-cleaved PARP (Asp214; Cell Signaling Technology), and rabbit anti-β-actin (clone 13E5; Cell Signaling Technology). Secondary antibodies: HRP-conjugated anti-rabbit IgG (Rockland Immunochemicals Inc., Pottstown, PA, USA). Bands were visualized using enhanced chemiluminescence (ECL; Thermo Fisher Scientific, Waltham, MA, USA) and densitometry was performed using iBright CL1500 imaging systems (Thermo Fisher Scientific).

**Table 1 ijms-26-10263-t001:** Composition of RIPA Buffer.

Reagents	Concentrations	Company
Tris trizma base	25 mM	Sigma-Aldrich
KCl	150 mM	FUJIFILM Wako Pure Chemical Corporation, Tokyo, Japan
EDTA-2Na	5 mM	DOJINDO LABORATORIES
NP-40 alternative	1%	FUJIFILM Wako Pure Chemical Corporation
sodium deoxycholate	0.5%	FUJIFILM Wako Pure Chemical Corporation
SDS	0.1%	FUJIFILM Wako Pure Chemical Corporation
Na_3_VO_4_	50 mM	FUJIFILM Wako Pure Chemical Corporation

### 3.7. Immunofluorescence

Cells were washed twice with 0.5% BSA/PBS and fixed with 4% paraformaldehyde (FUJIFILM Wako) for 15 min at RT. After two washes with 0.5% BSA/PBS, cells were permeabilized with 0.1% Triton X-100 (FUJIFILM Wako) for 6 min at RT, washed three times with 0.5% BSA/PBS, and blocked with 3% BSA/PBS for 1 h at RT. Following three washes, cells were incubated with NF-H primary antibody (1:400 in Solution 1; TOYOBO Co., Ltd., Osaka, Japan) for 1 h at 37 °C (or overnight at 4 °C) in the dark. After three washes, cells were incubated with secondary antibody (1:400 in Solution 2; TOYOBO) for 1.5 h at 37 °C in the dark, washed twice with 0.5% BSA/PBS, and then immersed in 1× PBS. Fluorescence images were acquired using a BZ-X710 microscope (KEYENCE, Osaka, Japan) with identical exposure settings across conditions.

### 3.8. Evaluation of Differentiated Neurons

Differentiation was assessed by immunostaining for the neuronal marker neurofilament heavy chain (NF-H; clone SMI-31; BioLegend, San Diego, CA, USA) and observation under fluorescence microscopy as described above.

### 3.9. In Vivo Tumor Xenograft Model and Treatment

All animal procedures complied with institutional and national guidelines and were approved by the Institutional Animal Care and Use Committee (IACUC) of Toyo University (Approval number: #2024-24) and Juntendo University School of Medicine (Approval number: #2023092). Female nude mice (BALB/c-nu, total *n* = 52) were purchased from The Jackson Laboratory Japan, Inc. (Yokohama, Japan). The mice were housed under a 12 h light/12 h dark cycle with free access to food and water and were allowed to acclimate for at least one week. The housing chamber temperature was maintained at approximately 23 °C.

HCT-116 cells (6 × 10^6^ cells in 100 μL HBSS with calcium and magnesium per mouse) were injected subcutaneously into female BALB/c nude mice (6 weeks old) to establish xenografts. After ~2 weeks, tumors from successfully engrafted mice were harvested and transplanted into a new cohort; this was repeated once more to expand cohorts, after which treatment was initiated 1 week later.

Mice were randomized to four groups: vehicle control, FOLFOX, EGT, or FOLFOX + EGT (*n* = 6). FOLFOX consisted of oxaliplatin (L-OHP, 6 mg/kg), leucovorin (LV, 90 mg/kg), and 5-fluorouracil (5-FU, 50 mg/kg), freshly prepared in sterile saline and administered intraperitoneally on a 1-day-on/3-days-off schedule. EGT (15 mg/kg in sterile distilled water) was administered by oral gavage on a 2-days-on/2-days-off schedule. All mice were euthanized for analysis at the end of week 3 of treatment. Tumor volume and body weight were recorded on days 0, 4, 8, 12, 15, 18 and 21 of treatment. We used female BALB/c nude mice to minimize confounders and ensure reliable measurements. Previous studies have reported sex-dependent differences in chemotherapy-induced neuropathy, including variations in pain sensitivity and peripheral nerve responses [47]. Based on these findings, we selected female mice in the present study to obtain consistent and interpretable outcomes in evaluating the protective effects of EGT on L-OHP-induced neurotoxicity; moreover, to improve internal validity for longitudinal endpoints, such as tumor volume and body weight.

### 3.10. Real-Time Quantitative PCR

Cells (SH-SY5Y, DLD-1, and HCT-116) were seeded at a density of 1 × 10^5^ cells/mL in 60 mm dishes (Iwaki & Co., Ltd., Tokyo, Japan) and incubated at 37 °C in a humidified atmosphere containing 5% CO_2_ for 24 h. Total RNA was extracted using the ReliaPrep RNA Cell Miniprep System (Promega K.K. Tokyo, Japan), and RNA concentrations were measured with a NanoDrop One spectrophotometer (Thermo Fisher Scientific K.K. Waltham, MA, USA). Complementary DNA (cDNA) was synthesized by reverse transcription using a 96-Well Thermal Cycler (Applied Biosystems. Inc. Waltham, MA, USA). Quantitative re-al-time PCR was performed on a Thermal Cycler Dice Real Time System III (Takara BIO Inc. Shiga, Japan) using SYBR Premix Ex Taq (Takara BIO Inc.). Primer sequences are listed in Table 2. The amplification protocol consisted of an initial denaturation at 95 °C for 1 min, followed by 40 cycles of 95 °C for 5 s and 60 °C for 30 s. The specificity of each reaction was confirmed by analysis of dissociation (melting) curves. The expression level of each gene was quantified relative to SH-SY5Y cells using the 2^−ΔΔCt^ method with *RPLP0* as an internal control. All experiments were performed in triplicate.

### 3.11. Statistical Analysis

Statistical analyses were performed in R (version 4.4.1). Group comparisons versus the control group were assessed using Dunnett’s test (multicomp package), two-sided. Data are presented as mean ± SEM unless otherwise indicated. A *p*-value < 0.05 was considered statistically significant.

Densitometric analysis of Western blot bands was performed using ImageJ software (Ver. 1.53e) (NIH, Bethesda, MD, USA) as previously described [48,49].

## 4. Conclusions

We demonstrated that ergothioneine (EGT) mitigates oxaliplatin-induced peripheral neuropathy (OIPN)–related neurotoxicity in human neuronal cells through anti-apoptotic and antioxidant actions without compromising oxaliplatin’s antitumor efficacy. EGT reduced oxidative stress, attenuated apoptosis, and preserved axonal/neurite integrity in both undifferentiated and differentiated neuronal cells. Across colorectal cancer models, EGT did not blunt oxaliplatin’s cytotoxicity in vitro or its tumor-suppressive effect in vivo. Moreover, in FOLFOX-treated mice, EGT coadministration mitigated chemotherapy-induced weight loss, indicating an additional supportive-care benefit. Collectively, these data provide preclinical evidence that EGT may serve as a useful adjuvant to oxaliplatin-based chemotherapy, offering protection against OIPN without loss of anticancer activity. Future studies should elucidate the molecular underpinnings of EGT’s effects in OIPN, define optimal dosing and scheduling, and evaluate clinical efficacy in patients receiving oxaliplatin-based treatment.

## Figures and Tables

**Figure 1 ijms-26-10263-f001:**
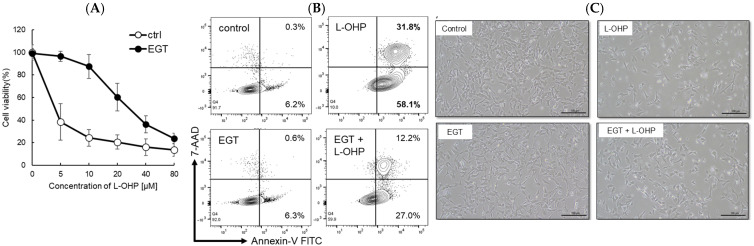
EGT protects against L-OHP–induced neuronal injury. (**A**) SH-SY5Y cells were pretreated with EGT for 48 h, followed by exposure to various concentrations of L-OHP and further incubation for 72 h. Cell viability was assessed using the SRB assay (*n* = 3; Welch’s *t*-test; *p* < 0.05). (**B**) Apoptosis of SH-SY5Y cells induced by L-OHP with or without EGT was evaluated by annexin V/7-AAD staining and analyzed by flow cytometry. (**C**) Representative phase-contrast images showing the morphology of SH-SY5Y cells treated with EGT (10 µg/mL) and L-OHP (8 µM).

**Figure 2 ijms-26-10263-f002:**
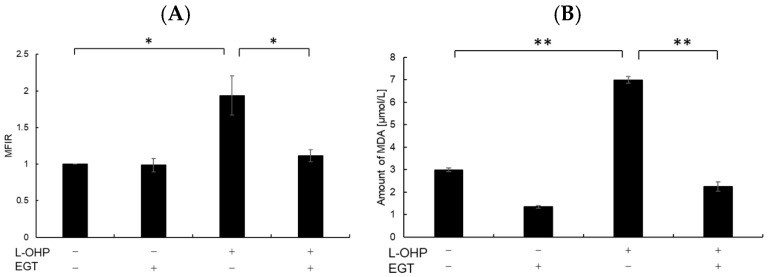
EGT reduces intracellular ROS and MDA levels induced by L-OHP. (**A**) SH-SY5Y cells were pretreated with EGT (10 µg/mL) for 48 h, followed by treatment with L-OHP (8 µM) for 24 h. ROS levels were measured according to the ROS assay kit protocol and analyzed by flow cytometry (Dunnett’s test; * *p* < 0.05). (**B**) Cells were pretreated with EGT (10 µg/mL) for 48 h, followed by treatment with L-OHP (8 µM) for 72 h. MDA levels were determined using the MDA assay kit and measured with a microplate reader (Dunnett’s test; * *p* < 0.05, ** *p* < 0.01).

**Figure 3 ijms-26-10263-f003:**
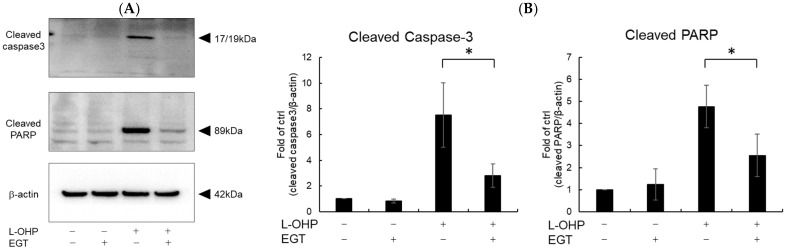
EGT inhibits L-OHP–induced apoptotic signaling. SH-SY5Y cells were seeded at a density of 5 × 10^5^ cells/mL and treated with EGT (15 µg/mL) for 48 h, followed by treatment with L-OHP (2 µM) for 24 h. After lysis with RIPA buffer, protein concentrations were determined by the BCA assay and adjusted to 4 mg/mL prior to Western blotting. (**A**) Protein expression of cleaved caspase-3, cleaved PARP, and β-actin was detected, and bands were visualized using an iBright imaging system. (**B**) Band intensities were quantified using ImageJ software (Ver. 1.53e; *n* = 3; Tukey’s test; * *p* < 0.05).

**Figure 4 ijms-26-10263-f004:**
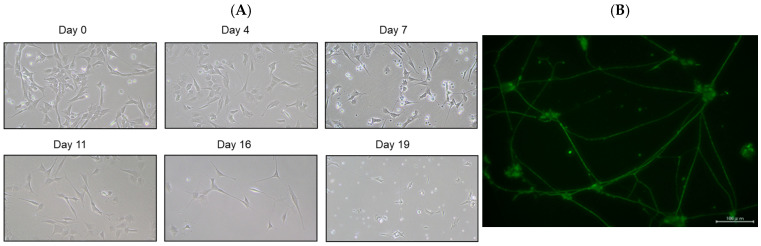
Neuronal differentiation of SH-SY5Y cells induced by retinoic acid. (**A**) Representative phase-contrast images showing the morphology of SH-SY5Y cells differentiated with retinoic acid at days 0, 4, 7, 11, 16, and 19. (**B**) Immunofluorescence staining of the neuronal marker NF-H on day 19 after differentiation.

**Figure 5 ijms-26-10263-f005:**
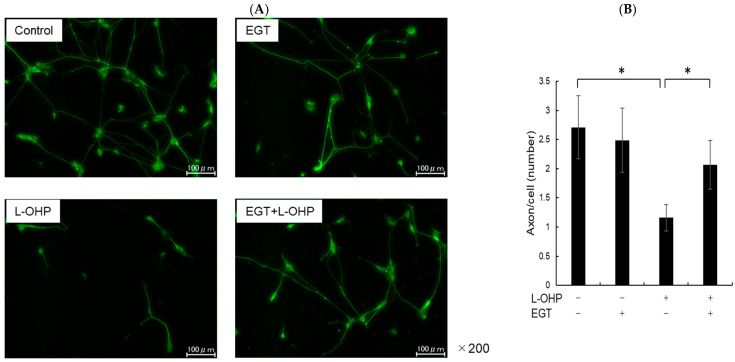
EGT suppresses L-OHP–induced neurotoxicity in mature neurons. (**A**) Mature SH-SY5Y neurons (day 19) were treated with L-OHP (2 µM) and/or EGT (20 µg/mL) for 48 h, followed by NF-H immunofluorescence staining and fluorescence microscopy. Image analysis was performed using ImageJ software (Ver. 1.53e). (**B**) Axon counts per cell were quantified. Images were obtained at three random fields per sample, and three independent experiments were performed. (Dunnett’s test; * *p* < 0.05).

**Figure 6 ijms-26-10263-f006:**
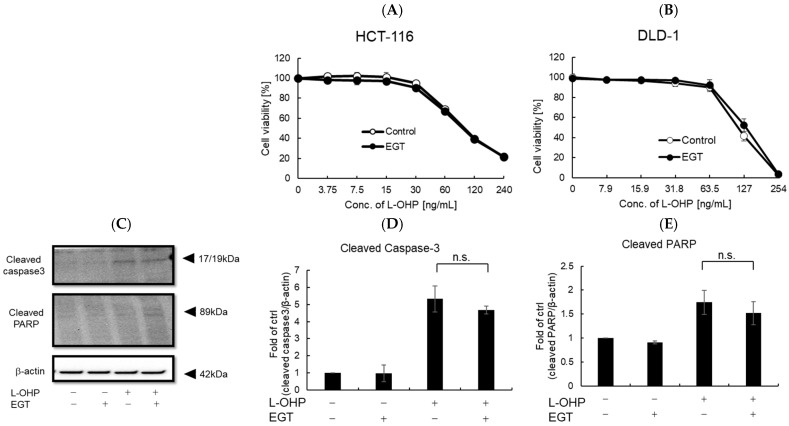
EGT does not interfere with the antitumor effect of L-OHP in vitro. (**A**,**B**) Antitumor effects of L-OHP with or without EGT were evaluated in HCT-116 and DLD-1 colorectal cancer cell lines. (**C**–**E**) Expression of apoptosis-related proteins (cleaved caspase-3 and cleaved PARP) was detected by Western blotting, and band intensities were quantified using ImageJ software (Ver. 1.53e; *n* = 3; Tukey’s test; n.s.: not significant).

**Figure 7 ijms-26-10263-f007:**
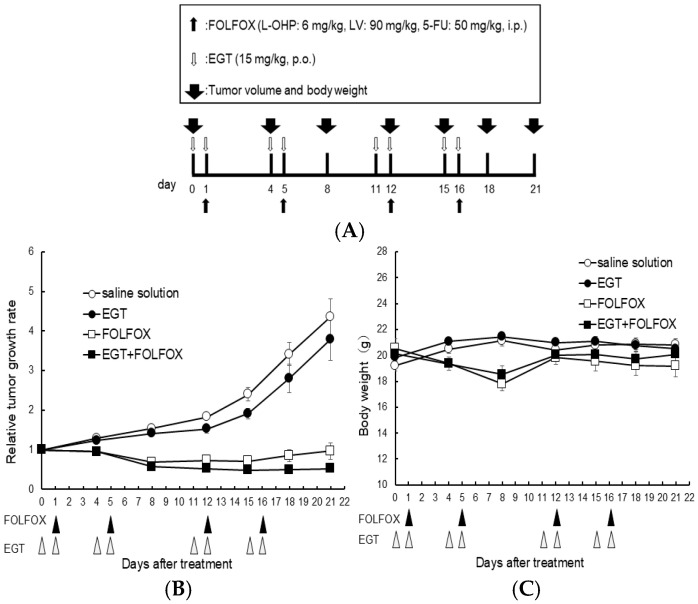
EGT does not affect the antitumor efficacy of FOLFOX chemotherapy in vivo. Nude mice were subcutaneously inoculated with HCT-116 cells and treated with EGT on days 7, 8, 11, 12, 18, 19, 22, and 23, and with FOLFOX (5-fluorouracil, leucovorin, and oxaliplatin) on days 8, 12, 19, and 23 after tumor implantation. (**A**) Dosing schedule was indicated, (**B**) Tumor volume and (**C**) body weight were measured on days 0, 4, 8, 12, 15, 18, and 21. Data are presented as mean ± SD (*n* = 6 per group).

**Figure 8 ijms-26-10263-f008:**
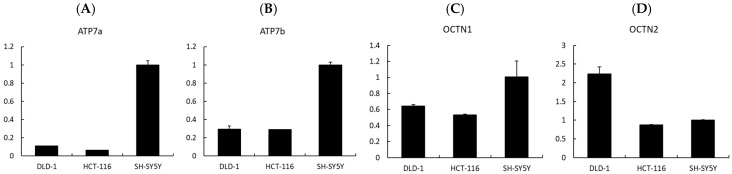
Expression analysis of transporters potentially involved in the neuroprotective effects of EGT. (**A**–**D**) Relative mRNA expression levels of *Atp7a*, *Atp7b*, *OCTN1*, and *OCTN2* were analyzed by quantitative real-time PCR. EGT appears to be associated with *Atp7a*, *Atp7b*, and *OCTN1*, whereas *OCTN2* may not be involved in neuroprotection. (Data are presented as mean ± SD, *n* = 3).

**Table 2 ijms-26-10263-t002:** Primers for Real-time quantitative PCR.

Gene	Pairs	Sequences
hRPLP0	Forward Reverse	5′-TGGTCATCCAGCAGGTGTTCGA-3′ 5′-ACAGACACTGGCAACATTGCGG-3′
hATP7a	Forward Reverse	5′-CCCTCTAGGAACAGCCATAACC-3′ 5′-ATACCACAGCCTGGCACAACCT-3′
hATP7b	Forward Reverse	5′-GGACCACAACATCATTCCAGGAC-3′ 5′-ATGAGCACGTCCATGTTGGCTG-3′
hOCTN1	Forward Reverse	5′-TGGACCTGTTCAGGACTCGGAA-3′ 5′-TAGGAGCATCCAGAGACAGAGC-3′
hOCTN2	Forward Reverse	5′-GCTACATGGTGCTGCCACTGTT-3′ 5′-CTGCCTCTTCAAATCGTCCCTG-3′

## Data Availability

The original contributions presented in this study are included in this article. Further inquiries can be directed to the corresponding author.

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
