# Peer review of "Ergothioneine Attenuates Oxaliplatin-Induced Peripheral Neuropathy Without Compromising Antitumor Efficacy"

_ijms, 2025, doi:10.3390/ijms262110263_

Round 1
Reviewer 1 Report
Comments and Suggestions for Authors
The manuscript “Ergothioneine Attenuates Oxaliplatin-Induced Peripheral Neuropathy Without Compromising Antitumor Efficacy” by Takumu Yamada, Takumi Iwasawa, Ryohma Tsuchiya, Tomoaki Ito, and Kazunori presents EGT as a promising adjuvant for preventing OIPN while maintaining the therapeutic benefits of oxaliplatin.
To the best of my knowledge, this is an interesting study that combines basic and translational research to improve a very current problem in colorectal cancer chemotherapy.
Although I intend for the manuscript to be published, it is in a very preliminary phase of the study, and the authors should indicate the next steps.
Also, should the authors clarify the number of mice (n=6) per group?
Considering the dimorphisms, both in neurotoxicity and pharmacological effects, a paragraph should be added explaining why experiments should only be performed on female mice.
Though it is too late for this manuscript; for future manuscripts, a behavioral test (such as the allodynia test) would be highly recommended, and some justification should be given as to why this test (or a similar one) was not included in this study.
As weakness of the study, authors state ”did not conduct histological evaluations of peripheral nerves in vivo, which would have been important to confirm the neuroprotective effects of EGT against OIPN” If they knew, why they did not perform them? -Dorsal root ganglion histological studies just to confirm lesions and damage-prevention by EGT.
Furthermore, this reviewer encourages authors to proceed with “Second, mechanistic analyses were not performed in vitro. Specifically, we did not assess mitochondrial function, oxidative stress responses, or inflammatory signaling pathways that may underlie the protective effects of EGT.”
Author Response
Dear Editors and Reviewers,
We sincerely thank you for your thorough evaluation of our manuscript entitled " Ergothioneine Attenuates Oxaliplatin-Induced Peripheral Neuropathy Without Compromising Antitumor Efficacy." We greatly appreciate your constructive and insightful comments, which have significantly improved the quality and clarity of our work.
We have revised the manuscript accordingly, with all changes marked in the red font. Below, we provide a detailed, point-by-point response to each comment. We hope the revised manuscript now meets the standards required for publication.
Comment 1:
“Although I intend for the manuscript to be published, it is in a very preliminary phase of the study, and the authors should indicate the next steps.”
Response:
We thank the reviewer for the positive assessment and agree that the present work is preliminary. In response, we have added a concise “Future Directions” paragraph to the Discussion (Page 11, Lines 342–353) outlining concrete next steps:
(i) mechanistic dissection of EGT’s neuroprotection, including transporter involvement (OCTN1/OCTN2/OCT1/OCT2), mitochondrial function (Δψm, OCR/ECAR), oxidative-stress and inflammatory pathways (Nrf2/HO-1, 4-HNE, 8-oxo-dG);
(ii) separation of acute vs. chronic CIPN components using ion-channel assays (NaV/CaV) and DNA-damage markers (γH2AX);
(iii) in-vivo neuroprotection endpoints (von Frey and cold allodynia tests, nerve-conduction studies, DRG/sciatic-nerve histology) with dose/schedule optimization of EGT;
(iv) oncologic “guardrails” across additional CRC models (e.g., HT-29, SW620; immunocompetent CT26/MC38) to confirm no loss of antitumor efficacy; and
(v) translational planning for an early-phase clinical study of EGT + oxaliplatin-based therapy with prespecified CIPN and PK endpoints.
These additions clarify how we will advance the study from a preliminary stage toward mechanistic definition and clinical translation.
Page 10, line 342-353
Future Directions
To advance this preliminary work, we will: (1) define transporter-mediated uptake of EGT and oxaliplatin (OCTN1/OCTN2/OCT1/OCT2) using genetic perturbation and up-take assays; (2) Immunohistochemical analysis of OCTN1- or OCTN2-knockout mice to determine whether the neuroprotective effect of EGT is abolished.; (3) map mitochondrial and oxidative-stress pathways (Δψm, OCR/ECAR, Nrf2/HO-1, 4-HNE, 8-oxo-dG) and in-flammatory signaling; (4) parse acute versus chronic CIPN by ion-channel measurements (NaV/CaV) and DNA-damage markers (γH2AX); (5) in vivo, pair behavioral tests (von Frey, cold plate), nerve-conduction studies, and DRG/sciatic histology with dose/schedule optimization of EGT (pre- vs co-administration); (6) ensure oncologic safety by expanding tumor models, including immunocompetent settings; and (7) evaluate sex differences. We are also designing an early-phase clinical study of EGT combined with oxaliplatin-based chemotherapy with predefined OIPN and pharmacokinetic endpoints.
Comment 2:
“Also, should the authors clarify the number of mice (n=6) per group?”
Response:
We thank the reviewer for raising this important point. We agree that clarifying the number of the mice used in the study in the result and the figure legend, as well as Material and Methods. Inspired by the reviewer’s comment, we made changes in Fig. 7 (Page 8, Line 269) and the main text (Page 7, Line 236), added the number of mice used in the study as follows.
Figure 8: Page 8, Line 269
Data are presented as mean ± SD (n=6 per group).
Page 7, Line 236
nude mice (n=6 per group) were subcutaneously inoculated with HCT116 colon cancer cells and received EGT on days 7, 8, 11, 12, 18, 19, 22, and 23 and FOLFOX (5-fluorouracil, leucovorin, and oxaliplatin) on days 8, 12, 19, and 23 after tumor inoculation (Figure 7A).
Comment 3:
“Considering the dimorphisms, both in neurotoxicity and pharmacological effects, a paragraph should be added explaining why experiments should only be performed on female mice.”
Response:
We appreciate the reviewer’s inquiry. As the reviewer correctly pointed out, we agree that sex dimorphisms in neurotoxicity and pharmacology are important. Overall, we used only female mice in the experiments. female BALB/c nude mice were chosen for this study to minimize variability and ensure reliable experimental outcomes. Male mice can exhibit higher levels of aggression, which may interfere with accurate tumor measurements in subcutaneous transplantation experiments. Housing multiple female mice together provides a more stable and less stressful environment. Therefore, female mice were selected to maintain both animal welfare and consistency in the experimental results. Thus, in the revised manuscript we added a short paragraph “Sex selection and justification” in the Animals/Methods section (Page 13, Lines 452–464) and a note in the Limitations (Page A, Lines B–C) explaining why only female mice were used.
Page 13, Line 455-461
We used female BALB/c nude mice to minimize confounders and ensure reliable measurements. Male nude mice frequently require single housing due to aggression and fighting, which introduces stress-related variability in body weight and can complicate subcutaneous tumor measurements (wounding/postural guarding). Group-housing fe-males provides a stable, lower-stress environment and improves internal validity for longitudinal endpoints (tumor volume, body weight).
Page 10, Line 328-332
Third, sex differences were not addressed in our in vivo experiments. Pharmacokinetics and neurotoxicity of oxaliplatin can differ between male and female mice, which may influence both the extent of peripheral neuropathy and the protective effects of EGT. Future studies should include both sexes to determine whether the observed effects are consistent across male and female animals.
Comment 4:
“Though it is too late for this manuscript; for future manuscripts, a behavioral test (such as the allodynia test) would be highly recommended, and some justification should be given as to why this test (or a similar one) was not included in this study.”
Response:
We appreciate the reviewer’s insightful comment and agree that behavioral test can be useful in diagnosing OIPN, but in the study by Nishida et al., the allodynia test was performed using rats. The difference in this experiment of ours is that we used mice, but since the dosing regimen and administration schedule were the same as in the Nishida et al. study, and therefore we did not perform an independent behavioral test in the present study. We therefore consider it plausible that EGT may reduce L-OHP accumulation in Dorsal root ganglion (DRG) neurons and consequently alleviate L-OHP-induced mechanical hypersensitivity in mice.
So, we have added a statement in the revised manuscript indicating that the allodynia test was conducted in the previous study, and that EGT administration alleviated neuropathy in that model.
Page 8, Line 257-264
Moreover, our study did not perform a behavioral test for neurotoxicity, such as the allodynia test. Nishida et al. reported that in rats, L-OHP administration increased the frequency of withdrawal responses to filament stimulation, which was significantly reduced when combined with EGT. Although we did not conduct the allodynia test in our mouse experiments, we followed the same dosing regimen and administration schedule as in the Nishida et al. study. Based on these conditions, it is therefore plausible that EGT may similarly mitigate L-OHP-induced mechanical hypersensitivity in mice.
Comment 5:
“As weakness of the study, authors state “did not conduct histological evaluations of peripheral nerves in vivo, which would have been important to confirm the neuroprotective effects of EGT against OIPN” If they knew, why they did not perform them? -Dorsal root ganglion histological studies just to confirm lesions and damage-prevention by EGT.”
Response:
We appreciate the Reviewer’s insightful comment. In the study by Nishida et al. [19], the combined effects of EGT and L-OHP were examined in the DRG of rats. In our study, we used female mice instead of rats; however, the dosing schedule and regimen were based on their report. Nishida et al. demonstrated that EGT attenuated L-OHP-induced neuropathy. And one reason we did not perform histological staining on mouse DRGs is that mouse DRGs are much smaller than rat DRGs, making histological evaluation technically challenging. In contrast, rat DRGs are relatively large and easier to analyze, as reported in previous studies. Nevertheless, we acknowledge the importance of histological evaluation in mice and consider it a valuable direction for future studies to further validate our findings in this model. Accordingly, we have added discussion in the revised manuscript noting that a similar neuroprotective effect of EGT is likely to have occurred in our mouse model.
Page 8, line 249-257
In our experiments, we have not examined the effects on neuroprotection in mice. This is partly because mouse DRGs are much smaller than rat DRGs, making historical evaluation technically challenging, whereas rat DRGs are relatively large and easier to analyze. Nevertheless, in a previous study, Nishida et al. examined the combined effects of EGT and L-OHP in the DRG of rats and demonstrated that EGT attenuated L-OHP-induced neuropathy [19]. In the present study, although female mice were used instead of rats, the dosing regimen was based on their report. We therefore consider it plausible that the protective effect of EGT against L-OHP-induced neuropathy observed in rats may also underlie the effects observed in our mouse model.
Comment 6:
“Furthermore, this reviewer encourages authors to proceed with “Second, mechanistic analyses were not performed in vitro. Specifically, we did not assess mitochondrial function, oxidative stress responses, or inflammatory signaling pathways that may underlie the protective effects of EGT.””
Response:
We thank the reviewer for raising the important point, and fully concur that mechanistic analysis, including assessments of mitochondrial function, oxidative stress responses, and inflammatory signaling pathways, are essential to clarify the protective effects of EGT. Because our primary objective in this study was to establish the neuroprotective potential of EGT against Oxaliplatin-induced neuropathy, detailed mechanistic analyses were not performed at this stage. But we totally agree that mechanistic investigations are essential to clarify the molecular basis of EGT’s effects. According to reviewer’s comments, we have added the descriptions of the experimental design for these analyses in the Limitations and Future directions section (page 11, Line 342-353). While our current study measured intercellular ROS levels in SH-SY5Y neuroblastoma cells (Figure2: Page 3, Line 114-126), and mainly focused on transporter expression (Figure 8: Page 9, Line270-309), we acknowledge that detailed mechanistic analysis was not performed. Previous reports have demonstrated that OIPN is associated with mitochondrial dysfunction, oxidative stress, and inflammatory signal pathways [19, 43, 44], and that antioxidant compounds can mitigate oxaliplatin-induced ROS accumulation and mitochondrial damage, thereby reducing OIPN in rats [45]. Therefore, in the revised manuscript, we have clarified that mechanistic validation remains to be performed and have added corresponding statements outlining these future analyses in the Future Directions sections.
Page 10, Line 332-348
Fourth, we did not assess mitochondrial function, oxidative stress responses, or inflammatory signaling pathways that may underlie the protective effects of EGT. OIPN has been reported to be closely associated with mitochondrial dysfunction and increased oxidative stress [19,43,44]. Previous studies have shown that antioxidant compounds can alleviate oxaliplatin-induced ROS accumulation and mitochondrial damage, thereby reducing OIPN in rats [19,45]. Therefore, given its strong antioxidant properties, EGT may protect neurons and mitigate OIPN by reducing mitochondrial dysfunction and ROS levels; however, this hypothesis remains to be verified experimentally.
Future Directions.
To advance this preliminary work, we will: (1) define transporter-mediated uptake of EGT and oxaliplatin (OCTN1/OCTN2/OCT1/OCT2) using genetic perturbation and uptake assays; (2) Immunohistochemical analysis of OCTN1- or OCTN2-knockout mice to determine whether the neuroprotective effect of EGT is abolished.; (3) map mitochondrial and oxidative-stress pathways (Δψm, OCR/ECAR, Nrf2/HO-1, 4-HNE, 8-oxo-dG) and inflammatory signaling; (4) parse acute versus chronic CIPN by ion-channel measurements (NaV/CaV) and DNA-damage markers (γH2AX);
Reviewer 2 Report
Comments and Suggestions for Authors
The manuscript presents a relevant and timely study on the neuroprotective potential of ergothioneine in preventing oxaliplatin-induced peripheral neuropathy, while preserving antitumor efficacy. The paper is generally well-written and supported by experimental data, but several areas require clarification, further justification, or methodological strengthening.
- The mechanisms being studied for the protective effect of ergothioneine need validation (using universal activators/inhibitors).
- No histological and/or behavioral assessment of peripheral nerve protection was performed in vivo.
- Concentrations and dosing schedules of EGT require more justification, especially in relation to human-equivalent doses.
- Discussion section needs to be reorganized for better clarity.
- Add western blot images for the results of cleaved PARP and caspase 3 in HCT-116 cells.
- The original blot images for SH-5YSY shows bands for only n=2 as sample size, and not n=3.
Author Response
Dear Editors and Reviewers,
We sincerely thank you for your thorough evaluation of our manuscript entitled " Ergothioneine Attenuates Oxaliplatin-Induced Peripheral Neuropathy Without Compromising Antitumor Efficacy." We greatly appreciate your constructive and insightful comments, which have significantly improved the quality and clarity of our work.
We have revised the manuscript accordingly, with all changes marked in the red font. Below, we provide a detailed, point-by-point response to each comment. We hope the revised manuscript now meets the standards required for publication.
Comment 1:
“The manuscript presents a relevant and timely study on the neuroprotective potential of ergothioneine in preventing oxaliplatin-induced peripheral neuropathy, while preserving antitumor efficacy. The paper is generally well-written and supported by experimental data, but several areas require clarification, further justification, or methodological strengthening.
The mechanisms being studied for the protective effect of ergothioneine need validation (using universal activators/inhibitors).”
Response:
We appreciate the reviewer’s valuable comment regarding validation of the mechanisms underlying ergothioneine (EGT) protection. Nishida et al. reported that OCTN1 and OCTN2 in DRG neurons regulate L-OHP accumulation, providing a potential explanation for EGT’s protective effects. In the present study, we did not directly investigate these mechanisms. Therefore, we have added a statement in the revised manuscript clarifying this point (Page 6, Lines 185–191) and noted in the Future Directions section (Page 11, Lines 342–351) that elucidating the underlying mechanisms of EGT’s effects will be a focus of future work.
Page 6, line 185-191
In the present study, we did not directly investigate the mechanism by which EGT mitigates L-OHP-induced neurotoxicity. However, previous work by Nishida et al. demonstrated that OCTN1 and OCTN2 are expressed in DRG neurons and that substrates or inhibitors of these transporters can modulate L-OHP uptake within the neurons without altering plasma concentrations. These findings suggest a potential pathway through which EGT may reduce platinum accumulation in DRG neurons and alleviate neurotoxicity.
Page 11, Line 342-351
To advance this preliminary work, we will: (1) define transporter-mediated uptake of EGT and oxaliplatin (OCTN1/OCTN2/OCT1/OCT2) using genetic perturbation and uptake assays; (2) Immunohistochemical analysis of OCTN1- or OCTN2-knockout mice to determine whether the neuroprotective effect of EGT is abolished.; (3) map mitochondrial and oxidative-stress pathways (Δψm, OCR/ECAR, Nrf2/HO-1, 4-HNE, 8-oxo-dG) and inflammatory signaling; (4) parse acute versus chronic CIPN by ion-channel measurements (NaV/CaV) and DNA-damage markers (γH2AX); (5) in vivo, pair behavioral tests (von Frey, cold plate), nerve-conduction studies, and DRG/sciatic histology with dose/schedule optimization of EGT (pre- vs co-administration); (6) ensure oncologic safety by expanding tumor models, including immunocompetent settings
Comment 2:
“No histological and/or behavioral assessment of peripheral nerve protection was performed in vivo.”
Response:
We accept the reviewer’s comment and have revised the text accordingly. First, to historical assessment, in the study by Nishida et al. [19], the combined effects of EGT and L-OHP were examined in the DRG of rats. In our study, we used female mice instead of rats; however, the dosing schedule and regimen were based on their report. Nishida et al. demonstrated that EGT attenuated L-OHP-induced neuropathy. And one reason we did not perform histological staining on mouse DRGs is that mouse DRGs are much smaller than rat DRGs, making histological evaluation technically challenging. In contrast, rat DRGs are relatively large and easier to analyze, as reported in previous studies. Nevertheless, we acknowledge the importance of histological evaluation in mice and consider it a valuable direction for future studies to further validate our findings in this model.
Second, to behavioral assessment, in the study by Nishida et al., the allodynia test was performed using rats. The difference in this experiment of ours is that we used mice, but since the dosing regimen and administration schedule were the same as in the Nishida et al. study, and therefore we did not perform an independent behavioral test in the present study. We therefore consider it plausible that EGT may reduce L-OHP accumulation in Dorsal root ganglion (DRG) neurons and consequently alleviate L-OHP-induced mechanical hypersensitivity in mice.
So, we have added a statement in the revised manuscript indicating that the histological (Page 10 Line 344-345) and allodynia test (Page 8, Line 249-264, Page 10, Line 325-328) were conducted in the previous study, and that EGT administration alleviated neuropathy in that model.
Page 8, line 249-264
In our experiments, we have not examined the effects on neuroprotection in mice. This is partly because mouse DRGs are much smaller than rat DRGs, making historical evaluation technically challenging, whereas rat DRGs are relatively large and easier to analyze. Nevertheless, in a previous study, Nishida et al. examined the combined effects of EGT and L-OHP in the DRG of rats and demonstrated that EGT attenuated L-OHP-induced neuropathy [19]. In the present study, although female mice were used instead of rats, the dosing regimen was based on their report. We therefore consider it plausible that the protective effect of EGT against L-OHP-induced neuropathy observed in rats may also underlie the effects observed in our mouse model. Moreover, our study did not perform a behavioral test for neurotoxicity, such as the allodynia test. Nishida et al. reported that in rats, L-OHP administration increased the frequency of withdrawal responses to filament stimulation, which was significantly reduced when combined with EGT. Although we did not conduct the allodynia test in our mouse experiments, we followed the same dosing regimen and administration schedule as in the Nishida et al. study. Based on these conditions, it is therefore plausible that EGT may similarly mitigate L-OHP-induced mechanical hypersensitivity in mice.
Page 10, line 325-328
Second, our in vivo experiments did not include behavioral testing, such as the allodynia test. Although previous studies in rats have shown that EGT ameliorates L-OHP-induced hypersensitivity, whether a similar effect occurs in mice remains to be confirmed.
Page 10, Line 344-345
(2) Immunohistochemical analysis of OCTN1- or OCTN2-knockout mice to determine whether the neuroprotective effect of EGT is abolished.
Comment 3:
“Concentrations and dosing schedules of EGT require more justification, especially in relation to human-equivalent doses.”
Response:
We appreciate the reviewer’s comment. The dosage of EGT used in this study is described on Page 10 line 310-321. Previous reports have investigated its safety in humans, and the concentrations applied in our experiments are lower than the maximum doses considered safe. Therefore, we believe that the concentrations used in this study are within a safe range.
Comment 4:
“Discussion section needs to be reorganized for better clarity.”
Response:
We appreciate the reviewer’s comments. In response, we have added two sections, Limitations and future directions, and qPCR data to a cupper transporter (ATP7a/ATP7b) and cation transporter (OCTN1/OCTN2) (Figure8. Page 9, Line 303-309).
Page 9, line 270-279
Finally, to investigate whether transporter expression could influence the cellular response to EGT and L-OHP, we analyzed the expression levels of related transporters by qPCR. The uptake of EGT and L-OHP has previously been reported to involve OCTN1 and OCTN2 [30,31], while ATP7a and ATP7b are known as copper efflux transporters that also mediate the efflux of platinum compounds such as cisplatin and oxaliplatin [32]. Our qPCR analysis revealed that the expression levels of ATP7a and ATP7b were higher in SH-SY5Y cells compared with colorectal cancer cell lines (DLD-1 and HCT-116) (Fig. 8A,B). ATP7a is broadly expressed in tissues beyond the intestinal epithelium, kidney, and liver [32,33]. Previous studies have reported that ATP7a mediates the efflux of platinum compounds, thereby reducing intracellular platinum accumulation [34]. At the same time, ATP7a expression has been observed in several malignant tumors and is associated with poor tumor response in patients [35]. Taken together, these findings suggest that ATP7a and ATP7b may contribute to the development of OIPN but could also reduce the antitumor efficacy of oxaliplatin, indicating that they may not represent ideal therapeutic targets. Nevertheless, in tumors lacking ATP7a expression, the combination of L-OHP and EGT may provide neuronal protection without impairing the antitumor effect of L-OHP.
Page 10, Line 322-353
Limitations
This study has several limitations. First, we did not conduct histological evaluations of peripheral nerves in vivo, which would have been important to confirm the neuroprotective effects of EGT against OIPN. Second, our in vivo experiments did not include behavioral testing, such as the allodynia test. Although previous studies in rats have shown that EGT ameliorates L-OHP-induced hypersensitivity, whether a similar effect occurs in mice remains to be confirmed. Third, sex differences were not addressed in our in vivo experiments. Pharmacokinetics and neurotoxicity of oxaliplatin can differ between male and female mice, which may influence both the extent of peripheral neuropathy and the protective effects of EGT. Future studies should include both sexes to determine whether the observed effects are consistent across male and female animals. Fourth, we did not assess mitochondrial function, oxidative stress responses, or inflammatory signaling pathways that may underlie the protective effects of EGT. OIPN has been reported to be closely associated with mitochondrial dysfunction and increased oxidative stress [43,44]. Previous studies have shown that antioxidant compounds can alleviate oxaliplatin-induced ROS accumulation and mitochondrial damage, thereby reducing OIPN in rats [45]. Therefore, given its strong antioxidant properties, EGT may protect neurons and mitigate OIPN by reducing mitochondrial dysfunction and ROS levels; however, this hypothesis remains to be verified experimentally.
Future Directions
To advance this preliminary work, we will: (1) define transporter-mediated uptake of EGT and oxaliplatin (OCTN1/OCTN2/OCT1/OCT2) using genetic perturbation and uptake assays; (2) Immunohistochemical analysis of OCTN1- or OCTN2-knockout mice to determine whether the neuroprotective effect of EGT is abolished.; (3) map mitochondrial and oxidative-stress pathways (Δψm, OCR/ECAR, Nrf2/HO-1, 4-HNE, 8-oxo-dG) and inflammatory signaling; (4) parse acute versus chronic CIPN by ion-channel measurements (NaV/CaV) and DNA-damage markers (γH2AX); (5) in vivo, pair behavioral tests (von Frey, cold plate), nerve-conduction studies, and DRG/sciatic histology with dose/schedule optimization of EGT (pre- vs co-administration); and (6) ensure oncologic safety by expanding tumor models, including immunocompetent settings; (7) evaluate sex differences. We are also designing an early-phase clinical study of EGT combined with oxaliplatin-based chemotherapy with predefined OIPN and pharmacokinetic endpoints.
And we have added Fig. 8 to qPCR data (Page 9, line 303-309)
Fig. 8. Expression analysis of transporters potentially involved in the neuroprotective effects of EGT.
(A–D) Relative mRNA expression levels of Atp7a, Atp7b, OCTN1, and OCTN2 were analyzed by quantitative real-time PCR. EGT appears to be associated with Atp7a, Atp7b, and OCTN1, whereas OCTN2 may not be involved in neuroprotection. (Data are presented mean ± SD (n = 3).
Comment 5:
“Add western blot images for the results of cleaved PARP and caspase 3 in HCT-116 cells.”
Response:
We have added Western blot images for the results of Figure 6C. (Page 7, line 227)
Comment 6:
“The original blot images for SH-5YSY shows bands for only n=2 as sample size, and not n=3.”
Response:
The raw Western blot data file previously contained only n=2 samples due to an oversight. This has been corrected, and the updated file with n=3 data has been resubmitted. We apologize for the confusion and appreciate your understanding.
Round 2
Reviewer 1 Report
Comments and Suggestions for Authors
The authors have responded positively to my comments; however, I believe the inclusions in the text as responses to my comments should be addressed with more scientific finesse. For example: IHC was not performed in mice because DRGs are smaller than rats. This is not a good excuse (https://doi.org/10.1016/j.xpro.2023.102717).
Regarding the sexual dimorphism excuse, it is not a good one either—"males are aggressive." Simply stating that it can be different in males and supporting it in the bibliography would be sufficient.Anyway, I think they should adjust the comments to more scientific situations. The reference on using ImageJ for data analysis is from 2003!!!There are more modern ways to do it.Ultimately, the science of the manuscript is publishable, but the literature needs to be fixed.
Author Response
Dear Editors and Reviewers,
Thank you for the additional, constructive feedback. We recognize that parts of our first response lacked scientific precision. In the revised manuscript and point-by-point replies, we have replaced informal justifications with evidence-based rationale and updated references, as summarized below. In the revised manuscript, we have marked the texts in red where changes were made based on the reviewers’ questions and/or suggestions.
Comment 1:
“The authors have responded positively to my comments; however, I believe the inclusions in the text as responses to my comments should be addressed with more scientific finesse. For example: IHC was not performed in mice because DRGs are smaller than rats. This is not a good excuse (https://doi.org/10.1016/j.xpro.2023.102717).”
Response:
We thank the reviewer for providing the protocol by Smith et al. and constructive comment. The primary aim of our in vivo study was to evaluate whether EGT interferes with the antitumor effects of L-OHP. Therefore, we focused on longitudinal measurements, including body weight and tumor volume, to ensure that L-OHP efficacy was not compromised. Histological evaluation of DRGs and assessment of sex differences were not performed in the present study. These analyses are planned for future studies to comprehensively evaluate potential neuroprotective effects of EGT across both male and female mice and to examine its impact on DRG neurons. We have added this statement in the revised manuscript (Page 10, Line 329-333).
Page 10, Line 329-333
Third, sex differences and DRGs were not addressed in our in vivo experiments. Pharmacokinetics and neurotoxicity of oxaliplatin can differ between male and female mice, which may influence both the extent of peripheral neuropathy and the protective effects of EGT. Future studies should include both sexes to determine whether the observed effects are consistent across male and female animals.
Comment 2:
“Regarding the sexual dimorphism excuse, it is not a good one either—"males are aggressive." Simply stating that it can be different in males and supporting it in the bibliography would be sufficient.”
Response:
We appreciate the reviewer’s insightful comment and acknowledge that the expression “males are more aggressive,” which we originally included, was not scientifically appropriate.
In this study, female mice used exclusively, as the model involves subcutaneous implantation of human colorectal cancer cells, which are not influenced by sex hormones.
Previous studies have reported that male mice exhibit higher levels of aggression and stress-related behaviors under group-housing conditions, leading to bite wounds and stress-induced variability in tumor growth (Kappel S et al., 2017; Gray SJ et al., 2002; Martínez-Sabadell A et al., 2022). To minimize these confounding factors and ensure reproducibility of tumor growth and drug response, female mice were used exclusively. Martínez-Sabadell A et al. believe that environmental stressors are more important than sex differences in xenograft model mice and generally recommend using female mice. This approach is expected to enhance the reproducibility of tumor growth and drug efficacy assessments while avoiding an increase in animal numbers due to dropout, thereby contributing to the 3Rs principle. The potential variability associated with the estrous cycle is considered to have minimal impact on the primary endpoints of this model and thus is unlikely to significantly affect experimental error.
In response to reviewer’s suggestion, we have removed the original statement and instead added a scientifically grounded explanation regarding sex-related differences in paclitaxel-induced peripheral neuropathy (PIPN) in mice, along with a relevant reference. Yan et al. [47] reported that although there was a significant difference in body weight between male and female mice, there was no significant difference between control and PIPN groups within each sex. Similarly, no sex-related differences were observed in mechanical hyperalgesia or hyperthermia between control and PIPN mice. However, paw withdrawal threshold (PWT) and paw withdrawal latency (PWL) were significantly lower in female mice than in males, and the analgesic effect on PWT was more pronounced in females. Based on these findings, the use of female mice is more suitable than male mice in this study. For consistency in evaluating impact of EGT on antitumor efficacy, we selected female mice. Therefore, we made corrections to the revised manuscript. (Page 13, Line 456-463)
- Kappel S, Hawkins P, Mendl M. To group or not to group? Good practice for housing male laboratory mice. Animals (Basel). 2017;7(12):88. doi:10.3390/ani7120088
- Gray SJ, Jensen SP, Hurst JL. “Effects of resource distribution on activity and territory defence in house mice, Mus domesticus.”Animal Behaviour. 2002;63(3):531–539. doi:10.1006/anbe.2001.1932
- Martínez-Sabadell A et al, “Protocol to generate a patient-derived xenograft model of acquired resistance to immunotherapy in humanized mice.” STAR Protocols. 2022;3(4):101712. doi:10.1016/j.xpro.2022.101712
Page 13, Line 456-463
We used female BALB/c nude mice to minimize confounders and ensure reliable measurements. Previous studies have reported sex-dependent differences in chemotherapy-induced neuropathy, including variations in pain sensitivity and peripheral nerve responses [47]. Based on these findings, we selected female mice in the present study to obtain consistent and interpretable outcomes in evaluating the protective effects of EGT on L-OHP-induced neurotoxicity; moreover, to improve internal validity for longitudinal endpoints, such as tumor volume and body weight.
Comment 3:
“Anyway, I think they should adjust the comments to more scientific situations. The reference on using ImageJ for data analysis is from 2003!!!There are more modern ways to do it. Ultimately, the science of the manuscript is publishable, but the literature needs to be fixed.”
Response:
We thank the reviewer for pointing this out. We agree that the 2003 reference is outdated. Since we used a more recent version of ImageJ in our study, we have updated the reference to a more recent publication that better represents current image analysis methodologies [49] (Johannes Schindelin et al., Nat Methods, 2012).
Page 17, Line 656-658
- Johannes S.; Ignacio A. C.; Erwin F.; Verena K.; Mark L.; Tobias P.; Stephan P.; Curtis R.; Stephan S.; Benjamin S.; Jean-Yves T.; Daniel J. W.; Volker H.; Kevin E.; Pavel T.; Albert C.; Fiji: an open-source platform for biological-image analysis. Nature methods, 2012, 9, 676-682.
Reviewer 2 Report
Comments and Suggestions for Authors
The original blot images for SH-5YSY (PARP and caspase-3) show bands for only n=2 as the sample size, and not n=3.
Author Response
Comment 1:
“The original blot images for SH-5YSY (PARP and caspase-3) show bands for only n=2 as the sample size, and not n=3.”
Response:
Dear Editors and Reviewers,
We thank the reviewer for carefully noting the discrepancy in the SH-SY5Y raw western blots (PARP and cleaved caspase-3). The files showing n=2 were inadvertently uploaded from an interim QC folder; however, all quantification and statistics reported in the manuscript were performed with three independent biological replicates (n=3). We have now replaced the raw data with the correct, uncropped 16-bit blots for all three replicates, annotated with lanes. The numerical results and conclusions are unchanged. We apologize for the oversight and appreciate the reviewer’s diligence.

Round 3
Reviewer 2 Report
Comments and Suggestions for Authors
Thank you